# Differential Diagnosis of Cyclic Vomiting and Periodic Headaches in a Child with Ventriculoperitoneal Shunt: Case Report of Chronic Shunt Overdrainage

**DOI:** 10.3390/children9030432

**Published:** 2022-03-18

**Authors:** Maximilian David Mauritz, Carola Hasan, Lutz Schreiber, Andreas Wegener-Panzer, Sylvia Barth, Boris Zernikow

**Affiliations:** 1Paediatric Palliative Care Centre, Children’s and Adolescents’ Hospital, Witten/Herdecke University, 45711 Datteln, Germany; c.hasan@kinderklinik-datteln.de (C.H.); b.zernikow@kinderklinik-datteln.de (B.Z.); 2Department of Children’s Pain Therapy and Paediatric Palliative Care, Faculty of Health, School of Medicine, Witten/Herdecke University, 58448 Witten, Germany; 3Department of Pediatric Neurosurgery, Klinikum Vest, Academic Teaching Hospital, Ruhr University Bochum, 45657 Recklinghausen, Germany; lutz.schreiber@klinikum-vest.de; 4Department of Radiology, Children’s and Adolescents’ Hospital, Witten/Herdecke University, 45711 Datteln, Germany; a.wegener-panzer@kinderklinik-datteln.de; 5Department of Pediatrics, Ostalb Klinikum Aalen, 73430 Aalen, Germany; sylvia.barth@kliniken-ostalb.de

**Keywords:** cyclic vomiting syndrome, periodic headache, shunt, overdrainage

## Abstract

Fourteen months after the implantation of a ventriculoperitoneal shunt catheter, a six-year-old boy developed recurrent, severe headaches and vomiting every three weeks. The attacks were of such severity that hospitalizations for analgesic and antiemetic therapies and intravenous rehydration and electrolyte substitution were repeatedly required. The patient was asymptomatic between the attacks. After an extensive diagnostic workup—including repeated magnetic resonance imaging (MRI) and neurosurgical examinations—common differential diagnoses, including shunt overdrainage, were ruled out. The patient was transferred to a specialized pediatric pain clinic with suspected cyclic vomiting syndrome (CVS). Despite intensive and in part experimental prophylactic and abortive pharmacological treatment, there was no improvement in his symptoms. Consecutive MRI studies reinvestigating the initially excluded shunt overdrainage indicated an overdrainage syndrome. Subsequently, the symptoms disappeared after disconnecting the shunt catheter. This case report shows that even if a patient meets CVS case definitions, other differential diagnoses must be carefully reconsidered to avoid fixation error.

## 1. Introduction

In the following case report, we describe a six-year-old male patient referred to our specialized pediatric pain clinic (SPPC) for further treatment of cyclic vomiting syndrome (CVS). By definition, CVS is a syndrome with episodes of uncontrollable vomiting separated by periods of relative well-being. Such episodes are often accompanied by other symptoms, including severe nausea, abdominal pain, headache, photophobia, phonophobia, and various autonomic symptoms similar to those seen in migraine [1,2]. The condition primarily affects children, but over the past decade, the recognition of CVS in adults has increased considerably [3]. A large proportion of patients require regular hospitalization for rescue therapies. The range of differential diagnoses is broad. Symptoms may be due to gastrointestinal disturbances (e.g., malrotation, gastroparesis), intracranial (e.g., intracranial masses, hydrocephalus) or abdominal (e.g., renal colic or ureteric obstructions) disorders, metabolic disturbances (e.g., fatty acid oxidation or urea cycle disorder, mitochondrial dysfunction), or drugs/toxins (e.g., cannabinoid hyperemesis syndrome) [4]. Comprehensive laboratory, radiographic, and endoscopic diagnostics detect a substantial proportion of these differential diagnoses. However, the diagnostic workup does not reveal seminal findings in most patients. The condition of these patients is referred to as CVS, which overlaps considerably with (abdominal) migraine [5]. Overall, there is a strong link between these two entities, as is reflected in the pharmacological treatment.

For the treatment of CVS, there are no evidence-based guidelines or controlled therapeutic trials, and treatment recommendations are mainly derived from expert opinion [5]. Experts recommend various therapeutic options for the treatment of CVS, although several therapy attempts are often necessary to achieve sufficient symptom control [4,5,6]. Therapeutic approaches are categorized as prophylactic, abortive, and rescue therapy. Like the preventive treatment for migraine, the antidepressant amitriptyline, anticonvulsants such as topiramate, and the beta-blocker propranolol are recommended in prophylaxis. For abortive therapy, antiemetics such as ondansetron and aprepitant as well as triptans are used. In rescue therapy during an attack, supportive medication with sedative and antiemetic drugs are used in addition to fluid therapy and the treatment of electrolyte disturbances. Monoclonal antibodies against the calcitonin gene-related peptide (CGRP) and the CGRP receptor are potential future therapeutic options, given the vital link between CVS and migraine [7,8].

Most case reports of pediatric patients with cyclic vomiting refer to otherwise healthy individuals [9]. In this case report, we present a patient with CVS who had a desmoplastic infantile ganglioglioma and was implanted with a ventriculoperitoneal shunt (VP shunt) catheter. The presence of a VP shunt expands the scope of differential diagnoses for periodic headaches and vomiting, even if the patient’s symptoms comply with the typical case definitions of CVS. Chronic shunt overdrainage may have a similar clinical presentation to CVS but is significantly underdiagnosed in patients with ventricular catheters. Shunt overdrainage, sometimes referred to as shunt-related headaches [10] or overdrainage syndrome [11], describes a condition in which an implanted ventricular catheter drains more CSF than is appropriate for the patient. Symptoms following overdrainage may manifest in an acute or chronic manner. Acute effects include subdural hematoma or posterior reversible encephalopathy. Chronic overdrainage syndrome has more subtle symptomatology with headaches, vomiting, or other neurological symptoms, which may only appear after a prolonged silent period [12]. To prevent overdrainage, current shunt systems include adjustable pressure valves and adjustable gravity valves. Although these technologies can significantly reduce, they may not entirely prevent a postural or gravity-generated siphon effect. This case report complies with the “CARE guidelines” [13].

## 2. Case Report

### 2.1. Patient Information

We report on a six-year-old male patient. He underwent surgery and aftercare for a desmoplastic infantile ganglioglioma in an external hospital. He was operated on twice; first in 2014 at six months of age in the right supratentorial region and again in 2016 due to tumor recurrence in the right precentral gyrus. In October 2018, a VP shunt (Miethke proGAV^®^ 2.0) had to be inserted due to a suspected hydrocephalus occlusus in follow-up magnetic resonance imaging (MRI) scans. There were no neurological symptoms. The catheter tip of the shunt was in a misplaced position in the right basal ganglia/right thalamus. In September 2019, the shunt opening pressure was adjusted from 8 to 10 cm H_2_O because MRI scans showed a narrowing of the lateral ventricles. Previously, vomiting had occurred once. At the time, no headaches were apparent. Unfortunately, after the surgery in 2014, he suffered from epileptic seizures, the most recent in 2019. Since 2014, he has received anticonvulsive therapy with oxcarbazepine.

After the brain tumor resections, he suffered from spastic hemiparesis of the left arm, mainly affecting the hand. He attended an inclusive kindergarten and, in 2020, was enrolled in a special school focusing on motoric impairment.

There was no family history of migraine, other primary headaches, gastrointestinal disorders, or cyclic vomiting.

### 2.2. Symptoms and Clinical Findings

In January 2020, the patient complained about the first episode of a severe bilateral headache in kindergarten. This attack lasted only for a few hours, and he had to vomit three times. The next attack occurred about four weeks later but lasted for a few days. During 2020, the pain attacks recurred about every three weeks and lasted approximately five to six days. The child would first vomit as soon as 15 min after the onset of the headache and then continue to throw up every 20 min for up to six days. He could not open his eyes, stand up, walk, or sleep undisturbed during this time. The intensity or occurrence of the headache was independent of body position; symptoms did not explicitly occur in the upright position or improve with the patient lying down. At the end of an attack, the boy suddenly felt well again within two hours and could eat normally. From the onset of the symptoms, extensive diagnostics were performed in an external regional hospital near the patient’s home as well as in an external tertiary-level center. These included laboratory diagnostics, an ophthalmological examination, an electroencephalogram, an electrocardiogram, echocardiography, sonographies of the abdomen and the shunt catheter, a lumbar puncture with cerebrospinal fluid (CSF) analysis, and MRI scans of the brain. These results were always unremarkable, and notably, the previously treating neurosurgeons ruled out shunt overdrainage based on the MRI findings and the atypical position-independent symptoms.

The patient had to be hospitalized repeatedly to treat severe electrolyte imbalances during attacks. There were several episodes of severe respiratory alkalosis at the beginning of the headaches, severe hypokalemia following metabolic alkalosis (hypochloremia), and one episode of seizure due to severe hypokalemia.

In July 2020, CVS was first suspected during neuropediatric rehabilitation and initial contact was made with our SPPC. The symptomatology met the North American Society for Pediatric Gastroenterology, Hepatology and Nutrition (NASPGHAN) [1], the Rome IV [2], and the International Classification of Headache Disorders version 3 (ICHD3) [14] criteria for CVS since no “other disorder” could be detected to explain the symptomatology. The classifications and diagnostic criteria are summarized in Table 1.

### 2.3. Therapeutic Intervention

The patient was hospitalized during the attacks. Several abortive measures were attempted in addition to the early intravenous fluid and electrolyte administration. The complications of respiratory alkalosis and hypokalemia did not recur with pre-emptive parenteral (infusion) therapy. Oral medication with oxcarbazepine was switched during the attacks to intravenous therapy with levetiracetam (and later to clonazepam) because of a seizure during a previous attack. Initial unsuccessful therapies were conservative abortive measures such as the administration of ibuprofen or acetaminophen to treat the headache, and dimenhydrinate or ondansetron to interrupt the vomiting. The escalation of analgesic medication to metamizole and piritramide also showed no decrease in the severity of the headaches. Under the hypothesis of CVS or abdominal migraine, a first prophylactic therapy with topiramate was attempted in June 2020. Due to ineffectiveness, this medication was discontinued. The prophylactic and abortive interventions are summarized in Table 2.

A subsequent abortive therapeutic trial with sumatriptan at the beginning of the headaches in August 2020 showed no effect. In August 2020, the patient initially presented to our outpatient SPPC. Prophylactic therapy with amitriptyline was started in September 2020. During the first inpatient stay in our SPPC in September 2020, zolmitriptan, aprepitant, dimenhydrinate, prednisolone, metamizole, levomepromazine, and esomeprazole were used during the attack. There was no significant effect on the symptom duration or severity under this regimen. During the next episode in October 2020, naratriptan and acetaminophen were used. The remaining abortive medication from the previous inpatient stay was unchanged. Again, there was no apparent effect on the symptoms. Prophylactic daily administration of aprepitant three days before the onset of attack also failed to show any effect. After the hospital stay, prophylactic therapy was extended with propranolol and coenzyme Q10. In November 2020, additional abortive medication with rizatriptan and nalbuphine was administered from the beginning of the episode. The first use of nalbuphine demonstrated a positive effect on the severity of the headache. After hospitalization in November 2020, the boy was implanted with a venous port-a-cath due to the need for regular intravenous therapy during the attacks of vomiting and headache. As of December 2020, prednisone, esomeprazole, levomepromazine, and nalbuphine were used during attacks. The prophylactic medication propranolol was discontinued in December 2020.

MRI scans in August 2020 showed a minimal increase in the lateral ventricular width compared to the previous scans and a constant midline shift of 3 mm. The tip of the VP shunt was still located in the right thalamus. However, the previous treating physicians assumed that this option was rather unlikely given the severe symptomatology, relatively unremarkable MRI findings, and repeated unremarkable external neurosurgical evaluations. The possibility of a VP shunt dysfunction and chronic overdrainage was discussed again with the parents in our SPPC early after the ineffectiveness of previous prophylactic drug interventions. The shared decision of parents and clinicians was to exhaust all pharmaceutical options prior to surgical intervention in the VP shunt. After the failure of classical prophylactics in the treatment of CVS and migraine, a trial of therapy with a monoclonal antibody against the CGRP receptor was discussed with the family, consistent with the Pediatric and Adolescent Headache special interest group of the American Headache Society recommendations for the treatment of refractory migraine [15]. The first trials with a subcutaneous administration of 35 mg erenumab (1.58 mg/kg body weight) were performed in March and April 2021. The administration was well tolerated with no side effects. Under this regimen, there was an increased frequency of attacks (3–4 per month) but a decrease in the duration (1–4 days). In May 2021, the family agreed to a final therapy trial with 70 mg of erenumab (3 mg/kg body weight). Again, there was no beneficial effect on the recurrent episodes of headache and vomiting. A timeline of interventions, symptoms, and prophylactic medications is given in Figure 1.

Further MRI scans of the brain in February 2021 showed a regression of the widening of the ventricular system and a slight increase in the midline shift. In May 2021, scans showed an accentuated overdrainage with an almost complete collapse of the anterior horn of the right lateral ventricle. Figure 2a shows the T2-weighted constructive interference in the steady-state (CISS) MRI sequence. The opening pressure of the shunt valve was set to 8 cm H_2_O at this time. On inspection, the valve could no longer be adjusted due to a malfunction. Due to the failure of therapeutic measures, overdrainage was reconsidered as the cause of the patient’s CVS, and surgical intervention was discussed with the family.

The VP shunt catheter was disconnected in June 2021. During surgery, unimpaired CSF flow was seen from the disconnected shunt despite the misplacement of the catheter tip. Additionally, the catheter was passed into a terminal Rickham reservoir to allow diagnostic puncture and pressure relief if needed. Immediately after the surgery, the last attack began with severe headaches and vomiting. This event lasted for four days. During this episode, therapy with levomepromazine and nalbuphine was given.

### 2.4. Follow-Up and Outcome

MRI scans shortly after the surgery showed a normalization of the width of the lateral ventricles, as expected, indicating a regression of the preexisting overdrainage. Figure 2b shows the T2-weighted CISS MRI sequence after catheter disconnection. Following disconnection of the shunt, there was no recurrence of attacks with headaches or vomiting. The boy developed a compulsive behavior pattern in the weeks following the procedure; we interpret this as an adjustment disorder after 18 months of severe illness. The obsessive-compulsive disorder was successfully treated during outpatient behavioral therapy in the following months. Follow-up MRI scans in February 2022 showed a constant unfolding of the ventricular system and a medialization of the midline. Subsequently, the patient remained free of symptoms. Explantation of the port-a-cath is planned in the following months.

## 3. Discussion

### 3.1. Overdrainage Syndrome

The different entities of shunt overdrainage complications can be categorized by their pathophysiology. Rekate et al. [10] distinguish five different categories: (1) “severe intracranial hypotension or low-pressure headaches” (sometimes called spinal headaches) which are associated with an increased siphoning effect of the shunt. The resulting decreased intracranial pressure (ICP) may cause the ventricles to collapse or, according to the Monro–Kellie doctrine, dilatation of the (venous) blood vessels, leading to venous congestion. When overdrainage leads to a ventricular collapse onto the catheter tip, the CSF may also become isolated from the main stream of circulation. This leads to (2) “intermittent ventricular catheter obstruction” (slit ventricle syndrome). The latter is also associated with at least intermittently increased ICP, as are the categories (3) “intracranial hypertension with small ventricles and a failed shunt” and (4) “intracranial hypertension with a working shunt”. Lastly, (5) “shunt-related migraine” describes headaches unrelated to the shunt. In some patients with shunt overdrainage and decreased ICP, stiff ventricular walls with low compliance prevent significant radiographic changes in ventricular width [16]. Additionally, premature suture fusion (suture sclerosis) and changes in the cranial vault or skull base may occur in infants, referred to as craniocerebral disproportion (CCD) [17]. Recent integrative models divide shunt overdrainage into intracranial hypotension (siphoning effect) and hypertension phenomena (following ventricular collapse/CSF isolation, acquired CCD, or venous hypertension). They also address the overlaps and transitions between overdrainage syndromes [12]. Several valve technologies are available to prevent overdrainage. Current shunt systems have adjustable pressure valves, adjustable gravitational valves, or a combination of both. Other systems include high-pressure valves, a membrane antisiphon, or flow-controlled devices. Such technologies can significantly reduce—but not completely prevent—postural overdrainage or the gravitational siphon effect [12,18,19]. If shunt overdrainage is detected, management should be adapted to the cause. Shunt optimization can be performed, for example, by adjusting the pressure level of the valve or changing the antisiphon device. A shunt transferal or a “shunt removal protocol” may be indicated for selected patients [12,20].

In the case of our patient, the Miethke proGAV^®^ 2.0 shunt system with Sprung reservoir was provided with a combined adjustable pressure valve and gravitational unit; a shunt prechamber was not implanted. We assume that the shunt overdrainage developed slowly. Initially, a single event of vomiting and radiologically detectable overdrainage occurred one year after shunt implantation, which led to an adjustment of the opening pressure of the adjustable valve in September 2019. After a silent period, recurrent severe headache and vomiting symptoms began in January 2020. Normal physical activity was possible for 2–3 weeks between the attacks. Increased ICP during physical activity (e.g., sports) resulted in overdrainage until the symptoms re-occurred. During the attacks, the ICP normalized again due to immobility and a recumbent position until the overdrainage developed again after the resumption of daily activities. This recurrent pattern resulted in the cyclic symptomatology. In the present case, the indication for shunt implantation can also be questioned retrospectively. From the medical history and clinical data available to us, there were no apparent signs of hydrocephalus at the time of shunt placement. Similarly, the fact that there were no clinical or radiological signs of increased ICP after disconnection of the shunt catheter argues against persistent hydrocephalus. The shunt as such functioned well, despite the misplacement of the catheter tip.

Among clinicians, the complication of shunt overdrainage is frequently underestimated. A recent survey among members of the American Society of Pediatric Neurosurgeons found that overdrainage was considered a complication in less than 15% of shunt patients, and that symptoms such as headache were likely due to other medical reasons [21]. However, there is a discrepancy with the actual rate of reported cases of overdrainage. There is a heterogeneous rate of actual reported cases of overdrainage ranging from 1% to 50% of all patients treated with a shunt, which may be explained by the inconsistent clinical criteria of overdrainage [12].

On the other hand, cyclical or repetitive symptoms such as headache, vomiting, and lethargy are repeatedly reported in cases of overdrainage [12,16,17,18,19,20,21,22]. Moreover, only some symptomatic patients show a radiographic correlation on MRI [19]. Hence, shunt overdrainage cannot be radiologically ruled out with certainty. In equivocal cases, ICP monitoring may be useful [12]. If shunt overdrainage is still precluded only due to regular ventricular width and only other differential diagnoses are considered, the risk of fixation failure may arise [23]. This error may continue when patients are referred to other practitioners or specialized clinics to manage other suspected medical diagnoses. This problem also arises because the CVS case definitions of the NASPGHAN, ICHD, and Rome IV criteria describe rather unspecific symptomatology that various underlying conditions can cause. This limitation generally applies to case definitions where there is no gold standard for a diagnosis, or the underlying pathology of the disorder is unknown or cannot be reliably diagnosed [24].

### 3.2. Pharmacological Treatment of CVS

For prophylactic and abortive therapy, medications with documented efficacy in CVS [3,4,5] were initially used in the case presented. Additionally, levomepromazine, an established neuroleptic used as an antiemetic in pediatric palliative care [25], was used during the attacks. We were aware of the polypragmatic approach of our treatment attempts; however, due to the severity of the boy’s symptoms, we felt an urgency to use this assortment of medications. After the failure of the previous medicinal approaches, other therapeutic approaches were discussed with the family. A subsequent therapy trial with the CGRP receptor antibody erenumab was attempted after critical consideration.

In the context of migraine headaches, CGRP is thought to play a central role in activating the trigeminovascular system. It is secreted along with substance P, neurokinin A, and pituitary adenylate cyclase-activating peptide (PACAP) after activation of the trigeminal ganglion. The strong vasodilatory effect of CGRP might be of importance in regulating cerebral blood flow in migraine. Based on the study findings that CGRP and PACAP can trigger migraine-like headaches, it has been considered that blocking CGRP or its receptor could treat or prevent the onset of an acute migraine attack. The injectable monoclonal antibody erenumab was approved in 2018 for the preventive treatment of episodic and chronic migraine in adults [26]. In the present case, we opted for an individual therapy trial after the family’s initial rejection of neurosurgical intervention, given the link between CVS and migraine and the reported favorable side effect profile of erenumab in adults and adolescents [27]. Likewise, our patient did not experience any side effects; however, there was no influence on the headaches and vomiting attacks due to the underlying shunt overdrainage. We interpret the increase in attack frequency as more severe overdrainage, later shown on MRI scans as an almost complete collapse of the lateral ventricles, rather than a side effect of CGRP receptor antibody therapy. The use of these novel treatments in migraine could still be effective in CVS but warrants further investigation [8].

### 3.3. Clinical Course

In the case of our patient, a neurosurgical intervention was discussed with our pediatric neurosurgical team early on should prophylactic therapy for CVS fail. With the narrowing of the lateral ventricles apparent on MRI scans in February 2021, a shunt transferal or disconnection surgery was recommended. At this point, however, the family still had high hopes for the pending therapy trial with erenumab and were hesitant about having another procedure after the previous surgeries. When symptoms did not improve under the erenumab therapy trial and the attacks became more frequent, we performed further MRI diagnostics. The scans finally revealed a notable overdrainage, and the family agreed to neurosurgical intervention. A shunt catheter disconnection was performed because the shunt valve could no longer be adjusted. The catheter was passed into a terminal Rickham reservoir, as no puncture option besides the borehole reservoir was previously available. Because of the uncertain initial indication for the VP shunt, we opted for a disconnection of the shunt catheter as a considerably smaller intervention rather than the reimplantation of a new valve or a shunt transferal. Immediately after surgery, the last brief attack occurred, and the patient was subsequently free of symptoms.

In retrospect, the patient probably could have been spared several months of severe recurrent symptoms had shunt overdrainage not been falsely dismissed so early. The current literature indicates that more attention needs to be paid to shunt overdrainage. The understanding of this condition is limited by the lack of generally accepted diagnostic criteria, the multitude of different treatment options, and limited comprehension of the relationships between the pathophysiologic mechanisms involved [12]. Current valve technology can prevent some of the complications. However, in the case of new-onset of definite or ambiguous neurologic symptoms, shunt overdrainage should always be considered, and appropriate measures such as ICP measurement, valve adjustment, shunt transferal, or shunt removal protocol should be taken. Cyclic vomiting syndrome is a diagnosis that should only be considered after all other possible causes have been excluded.

## 4. Conclusions

When pharmacological and interventional measures are ineffective with regard to a suspected diagnosis, differential diagnoses that were initially excluded must be reconsidered, especially in medically complex patients. In the case of recurrent symptoms such as headache, vomiting, or lethargy, the presence of overdrainage must always be considered in shunt patients, even if the radiological diagnosis appears to be unremarkable.

## Figures and Tables

**Figure 1 children-09-00432-f001:**
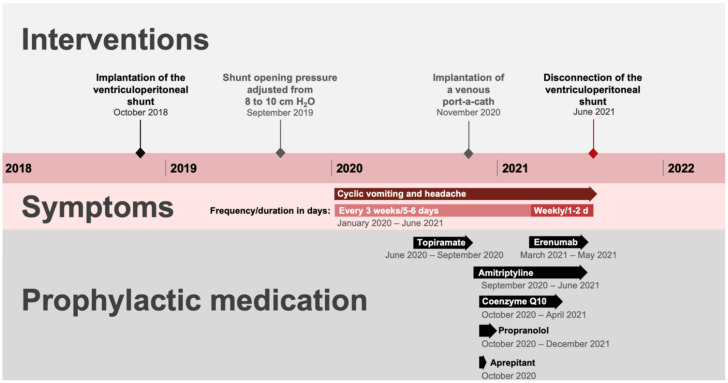
Timeline of interventions, symptoms, and prophylactic medications.

**Figure 2 children-09-00432-f002:**
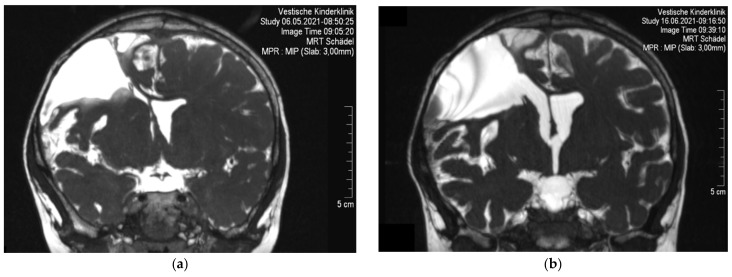
(**a**) T2-weighted CISS MRI sequence in coronal plane before disconnection of the VP shunt. Significant loss of volume of the lateral ventricles, with noticeable rightward distortion of the septum pellucidum and a craniocaudal diameter of the right lateral ventricular anterior horn of 2 mm as a sign of overdrainage. Right fronto-parietal appears a wedge-shaped lesion secondary to the ganglioglioma resection. (**b**) T2-weighted CISS MRI sequence in coronal plane after disconnection of the VP shunt. It shows unfolding of the ventricular system and medialization of the septum pellucidum with a widening of the anterior horn of the right lateral ventricle to 11 mm. The wedge-shaped lesion right fronto-parietal communicates with the right lateral ventricle.

**Table 1 children-09-00432-t001:** Current classification for the diagnosis of cyclic vomiting syndrome (CVS).

NASPGHAN [1]	Rome IV [2]	ICH-D-3 [11]
All the criteria must be met: 1. At least five attacks in any interval or a minimum of three attacks during a 6-month period; 2. Episodic attacks of intense nausea and vomiting lasting from 1 h to 10 days and occurring at least 1 week apart; 3. Stereotypical pattern and symptoms in the individual patient; 4. Vomiting during attacks occurs at least 4 times/hour for at least 1 h; 5. Return to baseline health between episodes; 6. Not attributed to another disorder.	**Children and Adolescents** Must include all of the following: 1. The occurrence of 2 or more periods of intense, unremitting nausea and paroxysmal vomiting, lasting hours to days within a 6-month period; 2. Episodes are stereotypical in each patient; 3. Episodes are separated by weeks to months with return to baseline health between episodes; 4. After appropriate medical evaluation, the symptoms cannot be attributed to another condition. **Neonates and toddlers** Must include all of the following: 1. Two or more periods of unremitting paroxysmal vomiting with or without retching, lasting hours to days within a 6-month period; 2. Episodes are stereotypical in each patient; 3. Episodes are separated by weeks to months with return to baseline health between episodes of vomiting.	A. At least five attacks of intense nausea and vomiting, fulfilling criteria B and C. B. Stereotypical in the individual patient and recurring with predictable periodicity. C. All of the following: 1. Nausea and vomiting occur at least four times per hour; 2. Attacks last ≥1 h and up to 10 days; 3. Attacks occur ≥1 week apart. D. Complete freedom from symptoms between attacks. E. Not attributed to another disorder (in particular, history and physical examination do not show signs of gastrointestinal disease).

ICHD-3: International Classification of Headache Disorders version 3, NASPGHAN: North American Society for Pediatric Gastroenterology, Hepatology and Nutrition.

**Table 2 children-09-00432-t002:** Prophylactic and abortive interventions.

Prophylactic Medication	Abortive Medication
Amitriptyline^po^ Aprepitant^po^ Coenzyme Q10^po^ Erenumab^sc^ Propranolol^po^ Topiramate^po^	Analgesics (acetaminophen^iv,po^, ibuprofen^po^, metamizole^iv,po^, nalbuphine^iv^, piritramide^iv^) Antiemetics (aprepitant^po^, dimenhydrinate^iv,pr^, granisetron^iv^, ondansetron^iv,po^) Corticosteroids (prednisolone^iv,po,pr^) Neuroleptics (levomepromazine^iv^) Proton pump inhibitors (esomeprazole^iv^) Triptans (naratriptanpo, rizatriptanpo, sumatriptannas, zolmitriptan^po^)

Medication in alphabetical order. Route of administration: ^iv^: intravenous, ^nas^: intranasal, ^po^: oral, ^pr^: rectal, ^sc^: subcutaneous.

## Data Availability

The data presented in this study are available on request from the corresponding author. The data are not publicly available due to privacy.

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
