# Peer review of "Differential Diagnosis of Cyclic Vomiting and Periodic Headaches in a Child with Ventriculoperitoneal Shunt: Case Report of Chronic Shunt Overdrainage"

_children, 2022, doi:10.3390/children9030432_

Round 1
Reviewer 1 Report
This is an interesting case report demonstrating that paediatric patients with severe and alarming symptoms of intracranial disease (headache, vomiting) can be misunderstood. When such symptoms are dramatic and episodic, but nevertheless require hospitalisation for pain relief and electrolyte corrections, the need for a correct diagnosis is obvious. In a patient with a low-grade brain tumour and ventriculoperitoneal shunt, shunt malfunction should be ruled out. In recent decades MRI has been helpful in the management of shunt patients. MRI and CT may give strong evidence of shunt malfunction (failure, over drainage, or unstable shunt function). MRI alone cannot, however, rule out shunt malfunction with certainty. In cases with dramatic symptoms, like the present one, ICP monitoring will demonstrate clear disturbancies with pathological pressure changes during attacks. This diagnostic procedure, will be a guideline to the neurosurgical management. The authors describe that the shunt was disconnected. I understand that the shunt was clamped or removed, and that the patient does not have a functioning shunt anymore. This should be made clear. The diagnosis of Cyclic Vomiting Syndrome seems challenging and management of such patients may be difficult. For a neurosurgeon, the use of such a diagnosis should perhaps be avoided in a patient like the present one, until shunt malfunction has been ruled out with certainty.
Author Response
Reviewer 1
This is an interesting case report demonstrating that paediatric patients with severe and alarming symptoms of intracranial disease (headache, vomiting) can be misunderstood. When such symptoms are dramatic and episodic, but nevertheless require hospitalisation for pain relief and electrolyte corrections, the need for a correct diagnosis is obvious. In a patient with a low-grade brain tumour and ventriculoperitoneal shunt, shunt malfunction should be ruled out. In recent decades MRI has been helpful in the management of shunt patients. MRI and CT may give strong evidence of shunt malfunction (failure, over drainage, or unstable shunt function). MRI alone cannot, however, rule out shunt malfunction with certainty. In cases with dramatic symptoms, like the present one, ICP monitoring will demonstrate clear disturbancies with pathological pressure changes during attacks. This diagnostic procedure, will be a guideline to the neurosurgical management. The authors describe that the shunt was disconnected. I understand that the shunt was clamped or removed, and that the patient does not have a functioning shunt anymore. This should be made clear. The diagnosis of Cyclic Vomiting Syndrome seems challenging and management of such patients may be difficult. For a neurosurgeon, the use of such a diagnosis should perhaps be avoided in a patient like the present one, until shunt malfunction has been ruled out with certainty.
Response:
Thank you for your positive feedback on our article. Overall, we have to emphasize that the patient was transferred to us from an external hospital (and the treating neurosurgical center) for the treatment of CVS. When the exclusion of overdrainage only on basis of MRI imaging was in question for us, a re-evaluation with our neurosurgical team was performed. This brought the differential diagnosis of overdrainage back into consideration and finally resulted in an adequate therapy. I hope this clarifies some of the issues that have been raised.
In our revision, we have highlighted the necessity of intracranial pressure measurement. The shunt was disconnected and passed into a terminal Rickham reservoir, as no puncture option was previously available. We have now described this in more detail.
Cyclic vomiting syndrome should be a diagnosis of exclusion. We have highlighted that, especially for such patients like the present one once again.

Reviewer 2 Report
Nicely presented manuscript, attempting to differentiate the underlying pathophysiology regarding a patient with VP shunt and clinical symptoms that could be attributed to shunt overdrainage.
A few remarks follow:
- As far as the introduction section is concerned, it is disproportionately divided, regarding the presentation of the Cyclic Vomiting Syndrome and the entity of shunt overdrainage. A more detailed presentation and definition of the latter entity is mandatory.
- There are some details that are missing regarding the patient’s medical history. Was the patient suffering preoperatively, or immediately postoperatively, after the first and second attempt at surgical excision of the lesion, from hydrocephalus? What was the MRI of the patient pre- and post-operatively (before shunt insertion)?
- You have mentioned that ‘The catheter tip of the shunt was located in the right basal ganglia/right thalamus.’ If this was the case, this is an inappropriate location.
- You mentioned that the ‘the shunt opening pressure was adjusted from 8 to 10 cm H2O because magnetic resonance imaging (MRI) showed narrowing of the lateral ventricles.’ I raise serious concerns that this practice is appropriate, based only on radiological findings.
- You do not provide details regarding the first episode of headache. That is, I am wondering if it was related with the adoption of an upright position (while the patient was previously lying) and what was the main area of pain distribution (i.e., frontal headache, retro-orbital, etc) and if it was relieved with the assumption of a supine position.
- You have mentioned that ‘treating neurosurgeons ruled out shunt overdrainage based on the MRI findings. Based on bibliographic research, I strongly consider that the diagnosis of shunt overdrainage in not established by MRI scans. Instead, it is mainly a clinical diagnosis and in very equivocal cases, it necessitates invasive ICP monitoring.
- You have mentioned that ‘most treating physicians assumed this option was rather unlikely in the face of severe symptomatology, relatively unremarkable MRI findings, and repeated unremarkable neurosurgical evaluations.’ According to literature reports, none of these criteria are capable of excluding the possibility of shunt overdrainage.
- You have mentioned that ‘On inspection, the valve could no longer be adjusted due to a malfunction.’ I would like to have a more detailed description regarding your interventions at this stage. For example, did you aspirate the chamber of the valve in order to ensure that there was a proximal shunt malfunction?
- It seems that this patient was relieved from his symptoms after shunt disconnection. Based on that, I am wondering if this could be an indication that he was never suffering from hydrocephalus or a transient post-operative hydrocephalus was established which subsequently subsided, and the patient was not shunt-dependent. I am looking forward to your comment.
- You have reported that ‘A shunt catheter disconnection was performed because the shunt valve could no longer be adjusted’. Another option could be the replacement of the central ventricular catheter and replacement of the valve mechanism with another one with adjustable valve opening (DP) pressure and anti-siphon device opening pressure (M-blue plus). I am looking forward to your comment.
Author Response
Reviewer 2
Nicely presented manuscript, attempting to differentiate the underlying pathophysiology regarding a patient with VP shunt and clinical symptoms that could be attributed to shunt overdrainage.
Response:
Thank you for your positive feedback on our article. Overall, we have to emphasize that the patient was transferred to us from an external hospital (and the treating neurosurgical center) for the treatment of CVS. When the exclusion of overdrainage only on the basis of MRI imaging was in question for us, a re-evaluation with our neurosurgical team was performed. This brought the differential diagnosis of overdrainage back into consideration and finally resulted in adequate therapy. I hope this clarifies some of the issues that have been raised.
A few remarks follow:
As far as the introduction section is concerned, it is disproportionately divided, regarding the presentation of the Cyclic Vomiting Syndrome and the entity of shunt overdrainage. A more detailed presentation and definition of the latter entity is mandatory.
Response:
Thank you for pointing this out. We had created the manuscript as the case had presented itself to us. Accordingly, we had not included the definition of overdrainage until later in the manuscript. However, since the symptoms described in our case report are finally diagnosed as overdrainage syndrome, we have now included the definition in the introduction.
There are some details that are missing regarding the patient’s medical history. Was the patient suffering preoperatively, or immediately postoperatively, after the first and second attempt at surgical excision of the lesion, from hydrocephalus? What was the MRI of the patient pre- and post-operatively (before shunt insertion)?
Response:
As previously stated, we were not involved in the initial treatment of the patient. Anamnestically, there were no clinical signs of hydrocephalus before implantation of the shunt catheter. According to the patient's mother, a follow-up examination was performed after the second resection of the tumor due to a delayed wound healing. In the MRI examination performed at that time, the treating team suspected hydrocephalus occlusus and indicated implantation.
We re-evaluated the MRI images we had from the previously treating hospital. After the first resection in 2014, there was a CSF leak postoperatively that spontaneously resolved. After the 2016 resection, there was a postoperative connection between the resection area and the right lateral ventricle, as seen in Figures 2 and 3. For us, there is no evidence of hydrocephalus in the available preoperative and postoperative images; there is also no indication of hydrocephalus in the images available to us before the VP shunt was placed. However, we must emphasize again that we were not involved in the patient's treatment at that time; we could only assess the MRI images.
You have mentioned that ‘The catheter tip of the shunt was located in the right basal ganglia/right thalamus.’ If this was the case, this is an inappropriate location.
Response:
The catheter position is obviously incorrect. We have highlighted this once again in the text. Here, however, we must also emphasize that we did not initially treat or operate on the patient.
You mentioned that the ‘the shunt opening pressure was adjusted from 8 to 10 cm H2O because magnetic resonance imaging (MRI) showed narrowing of the lateral ventricles.’ I raise serious concerns that this practice is appropriate, based only on radiological findings.
Response:
We interviewed the patient's mother about this again. She stated that the boy had previously experienced vomiting once without any other clear cause. We have included this information in the text.
You do not provide details regarding the first episode of headache. That is, I am wondering if it was related with the adoption of an upright position (while the patient was previously lying) and what was the main area of pain distribution (i.e., frontal headache, retro-orbital, etc) and if it was relieved with the assumption of a supine position.
Response:
The headache, as far as the then five-year-old patient could describe it, affected the whole head. The occurrence and intensity were position-independent. We have included this in the text.
You have mentioned that ‘treating neurosurgeons ruled out shunt overdrainage based on the MRI findings. Based on bibliographic research, I strongly consider that the diagnosis of shunt overdrainage in not established by MRI scans. Instead, it is mainly a clinical diagnosis and in very equivocal cases, it necessitates invasive ICP monitoring.
Response:
The diagnosis was ruled out by the previously treating neurosurgeons based on the MRI findings and the atypical position-independent clinic. For definite exclusion, an ICP measurement would certainly have been necessary. We have detailed this in the text.
You have mentioned that ‘most treating physicians assumed this option was rather unlikely in the face of severe symptomatology, relatively unremarkable MRI findings, and repeated unremarkable neurosurgical evaluations.’ According to literature reports, none of these criteria are capable of excluding the possibility of shunt overdrainage.
Response:
The diagnosis was ruled out by the previously treating neurosurgeons. During treatment at our specialized pediatric pain clinic, this differential diagnosis was revisited and discussed with the parents. We have pointed this out in the text.
You have mentioned that ‘On inspection, the valve could no longer be adjusted due to a malfunction.’ I would like to have a more detailed description regarding your interventions at this stage. For example, did you aspirate the chamber of the valve in order to ensure that there was a proximal shunt malfunction?
Response:
The valve was fixed and could not be adjusted. A shunt prechamber was not implanted. Therefore, unfortunately, we did not have this diagnostic option. We have added this to the text.
It seems that this patient was relieved from his symptoms after shunt disconnection. Based on that, I am wondering if this could be an indication that he was never suffering from hydrocephalus or a transient post-operative hydrocephalus was established which subsequently subsided, and the patient was not shunt-dependent. I am looking forward to your comment.
Response:
In retrospect, based on the medical history and clinical data, we assume that the patient never required a VP shunt. We have included this in the discussion section.
You have reported that ‘A shunt catheter disconnection was performed because the shunt valve could no longer be adjusted’. Another option could be the replacement of the central ventricular catheter and replacement of the valve mechanism with another one with adjustable valve opening (DP) pressure and anti-siphon device opening pressure (M-blue plus). I am looking forward to your comment.
Response:
According to the mother, there were no symptoms before the shunt was implanted. Based on the anamnestic data, we questioned the initial indication of the need for a shunt. Therefore, the choice fell on the smallest possible intervention with shunt disconnection. We have included this in the discussion section.
Round 2
Reviewer 2 Report
I studied the revised version of your manuscript, as well as your response to the suggestions that were proposed from my side.
I would like to state that, to the best of my knowledge, the pro GAV2.0 valvular mechanism has incorporated, by its own, a shunt prechamber. This is in contradistinction with the response that you have supplied to the Journal. Apart from that, the whole management of this patient is wrong, although not from your side. Ashunt was inserted that was necessary, a diagnosis of shunt overdrainage should be suggested, ans adequately treated from the treating neurosurgical team, and a non-working central shunt catheter was introduced, imaged on MRI, and remained untreated.
Author Response
Dear Reviewer,
Indeed, also to our surprise, the patient did not have a prechamber implanted. However, the patient has implanted a Sprung reservoir at the borehole, we added this information to the manuscript.
Secondly, the function of the shunt catheter was questioned. The catheter actually functioned very well, even too well in the context of overdrainage, despite the misplacement of the catheter tip. Accordingly, puncturing the reservoir would not have provided any additional information when the valve couldn’t be adjusted anymore. During surgery, unimpaired CSF flow was seen from the disconnected shunt. Therefore, we refrained from removing or replacing the shunt catheter at that point. We also added this information to the manuscript.
The questionable management of the patient is evident. This is why we submitted the article: Reevaluation of previously excluded diagnoses is always necessary, especially of overdrainage in shunt patients.
The most recent changes are highlighted in yellow.
Round 3
Reviewer 2 Report
I appreciate your reply to my comments. I consider that your revised version could be accepted for publication.